# Memorials as Healing Places: A Matrix for Bridging Material Design and Visitor Experience

**DOI:** 10.3390/ijerph19116711

**Published:** 2022-05-31

**Authors:** Brady Wagoner, Ignacio Brescó

**Affiliations:** 1Aalborg University, 9000 Aalborg, Denmark; 2Oslo New University College, 0456 Oslo, Norway; 3Departamento de Psicología Básica, Facultad de Psicología, Autonomous University of Madrid, 28049 Madrid, Spain; ignacio.bresco@uam.es

**Keywords:** design, experience, multimodal, engagement, movement, reflection, ritual, landscape

## Abstract

Memorials are increasingly used to encourage people to reflect on the past and work through both individual and collective wounds. While much has been written on the history, architectural forms and controversies surrounding memorials, surprisingly little has been done to explore how visitors experience and appropriate them. This paper aims to analyze how different material aspects of memorial design help to create engaging experiences for visitors. It outlines a matrix of ten interconnected dimensions for comparison: (1) use of the vertical and horizontal axis, (2) figurative and abstract representation, (3) spatial immersion and separation, (4) mobility, (5) multisensory qualities, (6) reflective surfaces, (7) names, (8) place of burial, (9) accommodating ritual, and (10) location and surroundings. With this outline, the paper hopes to provide social scientists and practitioners (e.g., architects, planners, curators, facilitators, guides) with a set of key points for reflection on existing and future memorials and possibilities for enhancing visitor engagement with them.

## 1. Introduction

Memorial sites aim to transform and contain painful experiences, including loss and trauma, within a recognizable symbol that is materially and spatially located. In this way, they function like gravestones, which come to represent a deceased loved one, providing a physical site to carry on continuing bonds [1,2]. But memorials are more complex in that they speak to a whole community’s painful experiences—*we* grieve for *our* dead. As such, they work to affirm and define a community, through creating and strengthening its memory for events in certain ways [3]. There is a complex tension here between the mourning of the individual dead and the collective dead, where a group’s values are put on display [4]. For example, traditional war memorials tend to transform individual deaths into a celebration of them as heroes that sacrificed for their country. Remembering is always combined with the forgetting of some events and aspects of remembered events that fall outside of the ‘social framework of memory’ [5]. For example, while considerable resources went into the memorialization of the WWI and WWII dead, the millions that were crippled in the wars were largely forgotten [6]. Memorial sites are thus spaces of shared memory that give different possibilities to express and interpret individual and collective loss. Such a process is not only about mourning but also can help individuals and societies to reinterpret the past, and in so doing, construct new orientations to the future [7]. These considerations lead us in this article to explore what features of memorials’ design afford (or constrain) different ways of engaging with them, which in turn allow for different sets of reflective and healing experiences.

Previous work on memorials as healing places has typically focused on their general social functions, with few empirical studies of visitors’ reactions. From studies in environmental design, Wasserman points out that memorial sites serve “intellectual, emotional, spiritual, and communal functions, including: (a) a place for memory, (b) a place for mourning, (c) a place for reflection and healing, (d) a place for ceremony, and (e) a place for collective action” (p. 44, [8]). From a heritage perspective, Viejo-Rose highlights “the primacy of the commemorative function of memorials as reminders of loss” (p. 466, [9]). While memorials’ representation of loss has been traditionally associated with the death and loss of human lives—whether caused by war, natural disasters or to a lesser extent pandemics—recent literature also includes environmental memorials commemorating biodiversity loss caused by climate change [10] or ‘survivor-memorials’ for those who lived through traumatic events [11].

Memorials’ potential healing functions have further been studied in relation to the notion of trauma from the perspective of psychoanalysis [12,13]. Within the so-called Psychoanalysis of the Cities approach, Minami and Davis [14] compare the repairing effects of a collective process of negotiation and symbol formation following the 9/11 attacks in New York and the atomic bomb in Hiroshima. In trying to express the “unspeakable” dimensions of collective trauma in the urban landscape, the process of psychosocial recovery in New York “reflects an ongoing and emotionally vested struggle for a more inclusive image and place identity for the city” (p. 56, [14]), whereas Hiroshima Peace Park memorial is marked by an overly idealistic symbolism of peace that has rendered a monolithic character for the city, thus silencing other possible ways of working through the traumatic past. Lastly, Watkins, Cole and Weidemann [15] analyze, in a longitudinal study, the healing effects of the Vietnam Veterans Memorial (VVM) on 62 male combat veterans diagnosed with posttraumatic stress disorder (PTSD). In focusing on the relationship between the memorial’s design features and the participants’ experiences, the authors conclude that the VVM acts as “a catalyst that allows a veteran to see, touch, remember, deal with, and master a loss he would rather have avoided or had difficulty expressing” (p. 368, [15]).

In this article, we understand healing in a broad sense to include both individuals and societies, and its work at affective, cognitive, and social levels. At the same time, we aim to go beyond the general claim of their healing potential to consider how specific features of memorial sites contribute to the visitor experience. While we recognize the importance of art history’s formal and iconographic categories of analysis in this endeavor, the present study is grounded in the memory and visitor studies literature, though we will mention the former categories where there is overlap. Our analysis hinges on the comparison between traditional and counter-memorials, which we will further elaborate on throughout this article. For now, suffice it to say that the former focuses on heroes and victories, subsuming death under a higher (usually national) cause. As a reaction against this, the latter focused instead on victims and loss, highlighting absences and leveling hierarchies. Thus, in contrast to the clear and fixed (intended) message of a traditional memorial, counter-memorials offer no unified, closed off and finalized account of the past. Defined in this way, the two seem to be exclusive opposites, but when we consider specific features of different sites, there can also be points of overlap. In connecting material design and visitor experience, we are not arguing that there is a kind of cause-effect relationship between the two, but rather that the former creates different fields of possibility for the latter. In this way, some memorials may end up promoting visitors’ appropriation of them in ways entirely unexpected by their architects. In what follows, we will present a matrix for considering the material dimensions of memorial sites and the ways in which they shape the visitor experience.

## 2. Materials and Methods

The primary data for this article comes from fieldwork conducted at three memorial sites: The Memorial to the Murdered Jews of Europe (MMJE) in Berlin, the National 9/11 Memorial (9/11) in New York City and the Valley of the Fallen (VF) in El Escorial, Spain (see Figure 1). These three sites were chosen because they were the most prominent and widely discussed at the time of the fieldwork and are large-scale installations that can be walked through which commemorate loss at national and international levels. Furthermore, MMJE and 9/11 are prototypical counter-memorials (and often discussed as such) whereas VF is a prototypical traditional memorial, a point we will substantiate throughout the article. While the focus of the analysis will be on their material and semiotic organization and how this is experienced, it is nonetheless important to understand the rather different historical events they commemorate and the circumstances under which they were built.

MMJE is Germany’s main Holocaust memorial, located near the Brandenburg Gate and Parliament in central Berlin. The architect, Peter Eisenmann [16], used an abstract, minimalist and non-didactic form of 2711 grey concrete stelae of uneven heights, arranged in rows with narrow aisles and descending floor. In a speech on the memorial’s inauguration in May 2005, he said his aim was to “begin a debate with the openness that is proposed by such a project, allowing future generations to draw their own conclusions. Not to direct them what to think, but to allow them to think.” The 9/11 memorial was constructed at ground zero and has as its main feature two memorial pools in the footprints of the twin tours, with the names of all the victims inscribed around the edges. Water, as a symbol of life and transition, cascades down the sides as well as into the center of each pool, which from the edge appears to be without a bottom. The architects Michael Arad and Peter Walker called their design ‘reflecting absence,’ which for them answered the question ‘how to articulate a void without filling it in?’ Finally, VF was built in the 1940s and 1950s by the then-dictator Francisco Franco in the wake of the Spanish civil war (1936–1939) as a forced symbol of national unity and reconciliation. The work was done by convicts, including former soldiers from the Republican side. It is situated on top of the Sierra de Guadarrama, near El Escorial, where the Spanish royalty is buried. Key features of the site are a monumental cross with a height of 150 m that can be seen from miles around, an esplanade and passageway, complete with a basilica and two prominent tombs. Given its history, VF remains a highly controversial site in Spain today.

At each of these sites we conducted observations of visitor’s behavior on-site (7 days at MMJE, 5 days at 9/11, and 3 days at VF), short interviews with visitors to the site (*N* = 45 at MMJE and 12 at VF), and more intensive data collection with a subjective camera (subcam), which records first-person video and audio (*N* = 5 subcam walks at MMJE, 8 at 9/11 and 5 at VF). The three methods provide a distinctive perspective on the site. The observations provided us with a general feel for the site and an idea of what people were doing there at different times of the day, while the short interviews (10–15 min) offered access to people’s varying interpretations and feelings about the site. The short interviews were done with any visitors to the site that were willing to talk with us, whereas for the subcam data-collection (which took around an hour) we recruited participants from our social network and local universities, where students were asked if they were interested in participating in research on memorial sites. Thus, the short interviews were done with a diversity of visitors to the sites (from a wide range of age groups and nationalities), whereas most participants doing the subcam task were in their 20s to 30s and from the country where the memorial was located. Triangulating the three methods did not reveal inconsistencies but allowed for different levels of breadth and depth of analysis.

The subcam was combined with either a walk-along interview [17] or a post-visit playback interview [18]. The idea was to get as close to participants’ streams of experience on-site and how it is constituted by various material aspects of the environment. Screenshots from the subcam videos will be used in this paper to illustrate features of memorials from the standpoint of visitors, and thereby highlight contextual and experiential qualities, powerfully captured through visual methods [19,20]. The interview is a necessary complement to the subcam data because not everything in a person’s visual field is registered or actively attended to as a meaningful component of their experience [21,22]. Furthermore, in human perception, what becomes a focal point of experience is often symbolically elaborated in reference to the cultural world to which the person belongs, as well as their personal history [23]. This signals a shift from the direct to the indirect (or symbolic) perception of the environment [24]. This transition can be identified in participants’ metaphorical associations, personal memories, and even sometimes embodied gestures. It highlights the importance of capturing not only representational qualities but also affective relating to the environment, which we take to be primary [23,25].

The principal dataset, encompassing the three memorial sites, was supplemented with a broader sample of sites that we have visited and conducted smaller-scale data collection (such as Hiroshima Peace Park and the Holocaust Names Memorial) or read about within the wider literature on memorial design and visitor engagement (esp. the Vietnam Veteran’s Memorial). This wider literature particularly contributes to the first step in our analysis: understanding the historical development of different dimensions of memorial sites, which then sets the frame for the characteristics of contemporary memorials and the experiences they channel. Using a broader set of examples can help to extend the generality and comparative aspects of the dimensions we focus on in our analysis. These were taken from prominent work done on the interpretation and use of memorials over the last twenty years.

## 3. Results

The results are organized around ten material dimensions of memorial sites. These were arrived at first by combining major themes that emerged in our open coding of data at each site—for example, the ambiguity in interpretations and norms due to the abstractness of MMJE’s design, sensory qualities of the water triggering reflection as well as the importance of the inscribed names of victims at 9/11, and the unease created by the vertical and figurative style of VF. After analyzing the material at each site, we began comparing them, especially with regard to the memorial/counter-memorial distinction, and in relation to a general review of historical developments in memorial forms found in the literature. Thus, the matrix we arrived at can be seen as a hybrid of empirical study and literature review. In this process, we created a longer list of dimensions, which were later subsumed under larger themes, as is typically done in coding procedures. This required balancing the need to create categories that were not too broad as to be overly vague, and not too specific so that they became useless for comparison. For each of the ten dimensions, we will briefly trace their historical emergence, describe their main material features, and how they are experienced and appropriated by different participants in our fieldwork and other studies in the area. The dimensions are interlinked and will thus be cross-referenced throughout.

### 3.1. Vertical and Horizontal

One of the most immediately perceivable material features of memorial sites is how their design elaborates on a vertical and/or horizontal dimension (cf. [26]). Traditional memorials tend to emphasize the vertical: Obelisks, such as the Washington Monument in D.C., the large cross at the Valley of the Fallen (see Figure 1) or statues on large pedestals are all typical examples. They can be seen from far away, and encourage visitors to stand at a distance, looking up at them in awe (see Section 3.4 below). This contrasts starkly with the horizontal axis of counter-memorials which cultivate a downward gaze amongst visitors and the immediate space of interaction. Historian Jay Winter [27] argues these features developed out of representations of Christ in the Western tradition, where the vertical expresses the hope of a better future in depictions of him on the cross and the horizontal expresses loss through depictions of his horizontally lying corpse. The First World War brought the horizontal axis into memorial architecture but in dialogue with the vertical axis. The horizontal is thus not an exclusive feature of counter-memorials, but they have developed it into one of their signature features.

Artist and architect Maya Lin described her design for the Vietnam Veterans Memorial (VVM) (Figure 2) as being like a “cut in the earth” that creates a “wound that is closed and healing”. (The same motif of a cut in the earth was similarly used in the original design for the Utøya memorial in Norway, where 77 members of the ‘Workers Youth League’ were killed by a right-wing extremist, utilized (Figure 2).) Lin’s design is entirely horizontal and requires the visitor to descend down into it. At the same time, one memorial wall visually connects with the Washington Monument, which creates a sharp contrast of forms (see further in Section 3.10 below). Similarly, MMJE features a descending floor on which a maze of unevenly sloping stelae are placed (Figure 1). Standing on the edge of the memorial, one can look over the whole thing, but when one is inside of it, the surroundings disappear and the visitor becomes disoriented. The uneasy feeling of being down inside the memorial was the experiential core of the visit and was where our participants’ interpretations tended to be generated. Our participants only felt that their visit was completed after they had gone into the memorial, become disoriented, ascended out and then looked over the memorial again to see it in its entirety (see [28] on ‘having an experience’).

At the National 9/11 memorial, visitors were primarily drawn to the memorial pools where their vision was guided downward. It was here that we found participants had their most personal and profound reflections on the events the memorial represents (see Section 3.6). Yet, while the memorial square is entirely flat, besides the trees and entrance to the museum, it is surrounded by the skyscrapers that define NYC. In particular, the One World Trade Center or ‘Freedom Tower’ was constructed right next to it and is the tallest building in the city, visually defining the skyline. Our participants did occasionally look up to it and say something like ‘there’s the Freedom tower’. Furthermore, on the anniversary of 9/11 (see Section 3.8) an art installation called *A Tribute in Light* projects two beams of light into the sky to create a spectral image of the Twin Towers. Similarly, those who lost loved ones on 9/11 also report looking up to where the towers stood while on-site and imagining the deceased up there [29]. The absence framed by the pools generates an imaginative vertical space of presence (see Section 3.6). In all this, we find a dialogue between horizontal and vertical forms, loss and hope, even at this prototypical counter-memorial.

### 3.2. Figurative and Abstract

The Second World War and events, such as Auschwitz, Hiroshima and Nagasaki exceeded the meaning and functionality of burial traditions typically based on conventional symbols and figurative representations of heroes and martyrs. The forms in which collective loss had been so far represented and socially remembered were called into question and became insufficient to memorialize events that felt to be unrepresentable and unthinkable [30]. Following the surge of abstraction in art decades earlier, memorials in the second half of the last century increasingly turned to non-representational and non-figurative motifs to address increasingly “controversial and uncertain attitudes toward history and its expression” (p. 38, [31]). Thus, while the art world became highly self-referential, abstraction in memorial forms spoke more to the difficulty in pluralistic democratic societies of finding sufficient consensus about the past to construct an unambiguous representation that all parties can agree on. Although there were earlier examples (e.g., the Deportation Martyrs Memorial in Paris, built in 1962), Lin’s VVM became paradigmatic for a new way of conceiving memorials based on the “conviction that neither the past nor its meanings are ever just one thing” (p. 14, [32]). Contrary to the notion of memorials as temples, where a monological (official) version of history is worshiped and revered, the abstract and minimalist form of counter-memorials was designed to enable different possible interpretations of the past. These sites then are meant to become forums capable of embracing disparate memories, feelings and interpretations.

In our fieldwork, we observed how the abstract and minimalist design of counter-memorials did seem to afford a more active engagement by our participants, leaving greater openness for experiencing and appropriating them. For example, MMJE’s architect Peter Eisenman intentionally cultivated an ambiguous form for the site, arguing that his goal was to “begin a debate with the openness that is proposed by such a project” [16]. The non-didactic design appears to encourage people to actively search for their own meaning memorial, much like someone might project images into an inkblot. In line with Dekel’s [33] findings, most of our participants made sense of the site by resorting to different metaphorical associations, for example, comparing the site to a graveyard, or interpreting the concrete stelae as obstacles that German Jews had to overcome [34]. This meaning-making process, afforded by the absence of a clear message, was realized through autobiographical memories at the 9/11 memorial. The visible absence framed by the two memorial pools at ground zero [35], invited many of our participants to fill these voids with different nostalgic memories associated with water spaces, which resulted in a personal appropriation of the site (see Section 3.6).

According to Stevens and Franck, contrary to straightforward memorials conveying a ready-made story, “absences require what German commentators often refer to as the ‘work’ of remembering, Erinnerungsarbeit” (p. 49, [31]). However, this can be a difficult task. As these authors go on to say, “for mourners and others who visit memorials, abstraction presents challenges” as “they refuse to offer easy and comforting answers” (p. 42, [31]), which representational memorials are often called to do. A figurative memorial of The Three Soldiers was added just across from Lin’s VVM wall memorial for this reason to appease certain veterans’ groups. Similarly, the sculptor of the Warsaw Ghetto Monument, Nathan Rapoport, “felt that carving human forms would emphasize the human loss and spiritual crisis created by the holocaust” (p. 52, [8]). Even at MMJE and 9/11, the museums located below the memorials communicate memory through a figurative form, using images, material objects and personal narratives. As such, one’s experience of the memorial is filled with specific, concrete images after having visited the museum.

### 3.3. Spatial Immersion and Separation

Memorials can be designed to be viewed from the outside or to spatially immerse visitors in them. At an experiential level, this tends to encourage the former ‘person–environment separateness’ (where the environment is felt to be distant, even alien, to the self) and the latter ‘person–environment mergence’ (where oneself is attuned to the environment) [36]. However, this is by no means a fixed relationship, as we will see below. It has already been discussed how vertical memorials can often create a sense of (as well as literal) distance between person and environment. While abstract designs have at least the potential to create an immersive space for visitors, Stevens and Franck [31] argue that the earliest spatially immersive memorial, Rodin’s Burgher of Calais (Figure 3), was entirely figurative. What made it innovative was that the figures were to be distributed across a square at ground level, such that visitors would almost run into them and come face to face with these anguished figures. (Against Rodin’s wishes, the figures were originally placed on pedestals in 1895; however, today they stand on a low basis in front of Calais’ new city hall). The memorial commemorates the French city’s leaders, who surrendered themselves to England, after an 11-month siege during the Hundred Years War, in exchange for sparing the local citizens. Rather than place heroically posed figures on pedestals (with the largest reserved for the eldest, de Saint Pierre), Rodin presents them without hierarchy at an everyday human level, together with the painful and fatalistic feelings that must have accompanied the moment.

Being thrown into a spatial design, rather than observing an object out there at a distance, helps to create a powerful atmosphere for visitors. As Böhme [37] explains, an atmosphere is neither in the environment nor the person but describes their embedded relationship. In Rodin’s design, we are affectively brought into the faithful moment of self-sacrifice through a down-to-earth encounter with the city’s leaders. In counter-memorials, this affective relating to the past is a principal feature. As we have already seen, visitors to MMJE get a feeling of unease, isolation and disorientation while inside the site, often without them having a clear idea of what it represents (see Figure 4). Yet, they somehow understand that this feeling is important to the site and are connected to it, as an eeriness that almost becomes sacred for some. Not unlike MMJE, the Nantes Memorial to the Abolition of Slavery gives one a feeling of being in the claustrophobic hull of a ship, as the slaves carried across the Atlantic once had to endure. The memorial is also located at the harbor in which slaves were brought into France (see Section 3.10). At the 9/11 memorial, all of our participants said they felt instantly calm and at ease when they stepped foot on the site, due to its park-like qualities of trees and sounds of water. But again, the immersiveness of the environment also connected them to ‘some of the darkness’ that the site represents (see Section 3.6).

Yet, it also happened that participants sometimes oscillated between immersion and separation. This was particularly clear in our fieldwork at VF. While the youngest participants that did the walk-along interview experienced it exclusively as a cold, distant and empty (associated with fascism), a woman in her 60s also saw it in terms of something warm, beautiful and familiar (associated with a Spanish church). On entering the first hall, she comments (cold associations are in **bold** and warm in *italics*):

“**It is a cold and empty hall, not transmitting any warmth**. […] Now I am approaching the gate. *Spanish ironwork screens have always been beautiful, and this one too*. **It’s a bit hard, it has bars**, *but I think it’s pretty. I also like the sculptures of the angels*, **also with hard angles**, *but beautiful. And the screen could be that of any cathedral, although more current. The basilica is also warmer than the outside*…”

In the next two rooms of the hallway her associations are almost entirely cold, but when she reaches the church area these shift into overwhelmingly warm affects: “I see benches, as if it were any church […] The site has a glitter that looks like a wedding just took place, as it could happen in any Spanish church […it] smells like a church, like celebration.” The warm familiar feeling lingers on as she approaches the grave of the founder of Spanish Fascism, J.A. Primo de Rivera, which she has affectively disconnected from the ideology it represents by its proximity to the familiar church.

### 3.4. Mobility

The key to the immersive memorial just described is the need to move through them. Since the end of the 19th century, memorials have provided visitors with more space to occupy within the site. This has given rise to what Michalski calls a ‘memorial complex,’ [38] in which the memorial becomes an entire setting comprising different commemorative elements. Memorial complexes following World War II are characterized by imposing spatial settings with “defined formal processional routes through them that choreograph particular movements, stopping points and views,” thus constraining “visitors’ attention, bodies and minds” (p. 112, [31]). Progressively, memorials have afforded more possibilities for visitors to explore and engage with them, framing them as actors rather than spectators. Again, a breakthrough within this tendency was VVM. Not only does the memorial require visitors to move through it rather than observe it from the outside, but visitors gain a major role as active participants, as they are no longer constrained by a ready-made script guiding their visit. As Lin remarks in her original proposal, “the memorial is composed not as an unchanging monument, but as a moving composition, to be understood as we move into and out of it” (p. 4, [32]). In other words, freely moving through memorials becomes key to the way in which they are experienced and made sense of by visitors.

Our fieldwork studies showed a stark contrast between traditional and counter-memorials in terms of the possibilities for movement they afford. As an example of a traditional memorial, we found that the architectural layout of the basilica in the VF—with its large nave leading to the main altar—powerfully guides the visit, giving rise through ‘body memory’ [39] to a conventional sequence of movements expected when visiting a church. The participants’ tour converged towards the funerary axis formed by Franco’s a Primo de Rivera’s tombs next to the main altar, taken as the visit’s terminating end point [40]. This example contrasts sharply with the open visiting possibilities afforded in MMJE and 9/11. As an approachable and permeable space, these memorials do not have any fixed end point, which affords visitors many options for moving into and around the memorial space, as our participants did at these sites. The visit thus becomes an open exploration process. Material affordances in the MMJE stimulated participants to actively appropriate the site through bodily movements, which become meaningful in themselves [31]. As mentioned above, one participant’s experience of this memorial as something to actively engage with to ‘overcome the obstacles’ and find your way through and out of the site, led him to associate the gradual built up of stelae with the obstacles Jews had to go through during the Nazi era [34]. Wasserman [8] points to a similar association with the VVM whereby visitors can experience the escalation of the war with their body movements. In her own words:

“Instead of lengthening the wall to fit all the war casualties, Maya Lin widens the wall as the death toll increases. This width is accommodated by digging down into the earth. Following the path into this depression gives the visitor a sense of the war’s expansion and the fear and despair that this entailed. Climbing out of the memorial offers a resolution, with the last name on the wall being the last person to die in the war”(p. 46, [8])

In sum, our results seem to be in line with other cases, empirically supporting the notion that interpretation of counter memorials is acted and felt rather than seen or read [31]. If we compare the VF with the MMJE, we can see how, in the latter case, the basilica has a more culturally recognizable functionality and meaning that constrains the type of actions one can expect in such a place (see Section 3.8). Conversely, the abstract design of the MMJE plays on the ambiguity of its meaning—explicitly intended by its architect—thus prompting a more active engagement with the possibilities and meanings the place offers. Paraphrasing Wasserman [8], we could say that in memorials such as the MMJE or the VVM, “the layers […] of understanding can expand as the observer/participant interacts with the memorial site”, thereby furnishing “a variety of levels on which to meditate, mourn, and heal” (p. 52–53, [8]).

### 3.5. Multisensory Qualities

As memorials afforded a more active role to visitors by encouraging exploration through body movements and feelings, the importance of the whole array of senses has grown in how people come to experience these sites. While architect Juhani Palasmaa [41] argues that design has generally suffered from an ocular bias, this is particularly true of traditional memorials. By contrast, counter memorials tend to invite multisensory exploration, expanding the qualities of physical engagement. In Stevens’ and Franck’s words, “recent memorials are not simply looked at […]. They also afford tactile, haptic, kinesthetic and sonorous experiences that potentially ‘touch’ people and ‘resonate with’ their understandings of the past” (p. 146, [31]). A great variety of elements in today’s memorials (such as water, mirror-finished materials, structures with different sizes and textures, and trees changing foliage with their regenerative symbolism, etc.) not only appeal to our distal senses (sound and sight), but also to our proximal ones, such as smell and particularly touch. These involve an active and immediate physical interaction with the environment [15].

In the fieldwork conducted at 9/11, we could observe the important role of water as a multisensory element. Through the go-along interviews, each participant guided the visit, while sharing with the researcher his or her impressions prompted by the site’s material and sensory elements, the flowing water being the most noticeable among them (see Section 3.6). Water was first experienced through its sound and only later did visitors get a visual of it once by the pool. One participant even commented on the smell: “the strong smell of chlorine always gets to me, as though, it is like, a cleansing.” Participants also touched the water in the basin below the metal surface with names cut into it. In stark contrast with the 9/11 memorial, none of our participants touched anything during their visit to the VF. Perceived as a symbol of power, meant to impose—or in Young’s words [32], to bring viewers into submission—few material or sensory elements of the basilica invited physical engagement with the site, resulting in feelings of coldness, distance and even alienation [40]. However, even here we observed some participants that connected with the site by making cross-modal metaphors, such as the woman quoted above, who said the church “smelled like… celebration” (see Section 3.3).

### 3.6. Reflective Surfaces

According to Wasserman [8], meditation and reflection are among the main expected functions of memorials intended to promote transformation and healing both in individuals and groups. To that end, many memorials use reflecting surfaces, such as polished stone or still pools of water “to facilitate meditative reflection, aiding the visitor in re-envisioning a person or event” (p. 54, [8]). The author alludes to the reflecting role of water in memorials, such as the Oklahoma City Memorial and Maya Lin’s Civil Rights Memorial in Alabama. She also mentions the sense of dual reflection—both physical and psychological—afforded by the polished black surface in the VVM expressed in the words of a grieving mother: “I visit the Memorial, I still feel sadness for the loss of the many young lives, but I can also reflect […] Today, I find there is comfort in Remembrance; healing in Reflection. I found it at the Wall.” (p. 54, [8]). Elaborating on the role of reflecting surfaces, Sturken [42] argues that the wall in the VVM acts as a screen in two senses: as a protective shield against a painful past (as in Freud’s ’screen memories’ which conceal by acting as a substitute) and as a surface upon which innumerable memories and interpretations are projected (like a screen in a cinema).

In our fieldwork at the 9/11 memorial [43], water stood out, not only as the dominant sensory element (Section 3.5) but also as a powerful symbol of life and transition that prompted numerous reflections and memories in most of our participants. This can be seen in the following words from one participant recorded while standing by the first pool (Figure 5):

“I’m just noticing how it has individual streams of water coming down the sides, instead of a solid sheet of water. And how the edge has a pool of water that reflects some and is very still. And that [making gesture indicating a waterfall] has a lot of movement. And now I’m seeing all the names written here [touches names]. I’m seeing ‘m56’. I don’t know what that means. I’m seeing a rainbow down there. That’s very pretty. [pause] And how the water all goes into that pit. Hmmm. It is kind of ominous in a way. Just like you can’t see the bottom [gestures downward again with hands] and it is kind of like the water just slowly goes over the edge. I think it is cool. I like it, but it has something [pause], to me it reflects some of the darkness that happened in addition to being a nice place. It reminds me of like the gravity of it all. It also reminds me of in my hometown there are some man-made lakes and they have, I don’t know what you call it, it is a drain basically, and it is called the waterhole. When I was young and the lakes were full the water would flow into it. It was always terrifying [laughs]. It has like little ropes that make like don’t swim or take your boat past this. It was like terrifying. What if someone somehow slipped down there, it would be awful. You’d see somebody in a little tin boat just outside the buoys and think oh my god.”(pp. 228–229, [43]).

The participant transitions from a description of the sensory features of the memorial to an interpretation of what they stand for (e.g., ‘it reflects some of the darkness that happened’). Ecological psychology would describe this as a movement from direct to indirect perception of the environment, from sensory to symbolic qualities [24]. This transition is accompanied by visually attending to different elements of the pool (e.g., the cascading water, a rainbow and the pit) and by the movements of her hands imitating the water’s trajectory. There is an interesting moment when her hand seems to go down into the pit that she sees as bottomless (Figure 4). It is at this point, that the participant begins to engage with the ‘darkness’ of the place through memories from her childhood, which mixes nostalgia with fear. This was by no means a singularity within our sample. The memorial pools afforded different kinds of experiential associations with memories of water spaces from the participants’ past. In short, the presence of water helped create a reflecting space for the collective and individual memory to coalesce.

### 3.7. Names

The inscription of names on memorials radically changes their character, as it provides a connecting link to concrete individuals that have died. In traditional memorials, names were reserved for only significant individuals that can stand in for a nation or cause. For example, at VF, only the names of the Spanish dictator Francisco Franco and the founder of Spanish fascism José Antonio Primo de Ribera are present together with their bodies (see Section 3.9). (Franco’s remains were removed from the basilica on 24 October 2018. Therefore, his tomb and his name are no longer visible, although they were when our fieldwork was carried out.) The whole visit is orchestrated to culminate with their tombs at the end of the passageway, just after the chapel (see Section 3.3 and Section 3.4). The more than thirty-three thousand ex-combatants buried on-site (mostly from Franco’s side but also the Republicans) are only mentioned indirectly as a mass subject in the phrase ‘those that died for God and Spain’—an absence our participants tended to notice. (The dead as a mass subject can also be seen in memorials to the unknown soldier, where the nameless can come to represent the entire nation). This motto was repeatedly used by Franco’s National Catholic ideology, such that it implies only those that died on Franco’s side are really being acknowledged. The centrality of the two tombs and peripheral mention of the Republican dead creates a clear ‘funerary hierarchy’ [44] (see Section 3.9).

In contrast to VF, counter-memorials aim to avoid creating hierarchies around names, preferring to dwell on individual losses without attaching them to any higher cause. Thus, they aim to create differentiation from a mass subject. The practice of listing the names of the individual dead began as a local memorial practice in the US after the Civil War, in Australia after the Boer War in 1903 and in England after WWI [31]. It has since become a key feature of many counter-memorials, beginning with VVM (Figure 6). Lin herself said “These names, seemingly infinite in number, convey the sense of overwhelming numbers, while unifying these individuals into a whole … Brought to a sharp awareness of such a loss it is up to each individual to resolve or come to terms with this loss. For death is in the end a private matter, and the area contained within this memorial is meant for private reckoning” (see the documentary film *A Strong Clear Vision.*). The memorial presents the names chronologically according to the day of their death, which created some controversy, especially in that it presents the progression of the war in terms of the number and scale of individual dead.

The 9/11 memorial, by contrast, organizes names of the dead around the location of death (e.g., in the North or South Tower, or airplane flight), group affiliation (e.g., company, firefighter and police divisions) and ‘meaningful adjacency,’ where individuals sharing a personal, work or family relationship were placed close together (Figure 6). Located on the edge of the immense memorial pools, the names create a tension of scale and a distinctive way of looking [35]. Whereas the pools encourage a reflection on the event as a whole—through the physical void and the conglomerate of dead it created (see Section 3.6)— “names affirm the individual out of the mass subject of a disaster, asking us to reflect on each life lost as a unique entity” (p. 319, [35]). Many of our participants adopted a practice of looking for names similar to their own or someone they knew, and pausing in front of them to imagine who they were, and what their life and relationships were like. One participant who would say a prayer next to familiar-sounding names commented:

“I find myself looking at the names to see if perhaps there was someone I knew. It’s interesting [pause]. I don’t even know why that would be an actual desire [pause]. Maybe it is a desire to feel more connected to them. I’ve seen names that look familiar but no one that I knew. And also when my friends visit they look to see if ‘oh that person has my first name,’ ‘oh that person has my last name’. I think people are just trying to find some connection. It’s somewhat unconscious.”

Finally, at the newly opened Holocaust Names Memorial, the 130,000 individuals from the Netherlands that died in the concentration camps are listed alphabetically, together with their birthday and age at which they died. This allows visitors to search for particular names more easily and see family connections—for example, Anne Frank and her family. Similar to what we found at 9/11, one visitor we spoke to searched for his own name at the site, and on finding it, he did historical research on who the person was and their life trajectory, discovering some parallels to his own family migration history. This is again a strategy of building a personal connection to the unique individual represented by a name. It contrasts sharply with MMJE, where visitors struggled to find a point of personal connection with the abstract site, with few indicators as to what it stands for.

### 3.8. Accommodating Rituals

Rituals can be seen as routine behavior with a high symbolic load [45]. This symbolic load carries important identity functions for groups, which in the face of loss, helps to reaffirm the bonds that hold them together [46]. Rituals are also heavily constrained by normative expectations and the psychological habits they form, being characterized by less free-play than other forms of behavior [47]. They range from being personal (performed individually, such as the practices around the names) to collectively shared rituals (Durkheim’s approach). At memorial sites, people often adopt ritual behavior akin to visiting a cemetery, such as leaving flowers, stones and other memorabilia. This can help visitors feel connected to the site and with some agency in the face of tragedy. At the Children’s Peace Monument in the Hiroshima Memorial Peace Park, there is a statue dedicated to a little girl who died of radiation poisoning. Schoolchildren make thousands of origami cranes which are placed on the statue as a symbol of peace and hope, renewing community memory and commitments to the cause [8]. Names (Section 3.7) provide a particularly fitting loci for placing such objects and, as such, are powerful components of ritual acts. At 9/11, the names are cut into the metal, which leaves a space to situate objects. The site foundation has institutionalized the practice of leaving a white rose on the names of those having a birthday (Figure 5). At VVM and the Holocaust Names memorial, objects are left on the ground at the edge of the memorial (Figure 5). The latter is particularly interesting in that the planters at the memorial are filled with round white rocks, which encourages visitors to practice the Jewish funerary ritual of leaving a stone on a grave (see Section 3.4). This creates a powerful visual effect that in turn prompts other visitors to do the same. While ritual itself is not a material dimension, memorials can design features that encourage the enactment of different kinds of rituals. As Wasserman points out:

“Key components of memorial spaces relate specifically to the ritual use of space in memorial practices. There is a need for those who have incurred a loss to search for meaning, to grieve, to contemplate and reflect on the person or event, and, at times, to use the memorial as a place to catalyze change. Many successful memorials contain these ingredients through the symbolic and functional use of siting, materials, and spatial organization”(p. 60, [8]).

However, memorials accommodate rituals and commemorative events (such as on the anniversaries of the tragic events) in different ways, and with greater and lesser ease [31]. Memorials consisting of open spaces, such as the 9/11 or Hiroshima Peace Park Memorial, can more easily host collective commemorations that help to reinforce common remembrance of the tragic event as well as community bonds. Other memorials, such as VF, seem tailor-made to host more traditional and predictable ceremonies. Commemorative practices in the memorial’s basilica—masses for the fallen celebrated by Dominicans—are consistent with the memorial’s architectural space and the National Catholic political ideology that inspired its construction. (Until recently, each November 20, coinciding with Primo de Rivera’s and Franco’s death anniversary, the VF has been hosting celebratory rituals by those nostalgic for the dictatorship with Fascist anthems and Roman salutes.) In contrast, the design of memorials such as the MMJE—with its 2711 concrete stelae arranged in narrow aisles—cannot accommodate these kinds of formalized and collective practices, as there is no place large enough for even a small group to stand together inside the memorial [31]. The behaviors that happen here do not have the heavily scripted character of formal rituals and, in many ways, are their opposite.

In fact, the ongoing discussions about the appropriateness and meaning of MMJE are part of what continues to animate it as a site of memory—as different perspectives are continually woven into the meaning people give to the place [7]. In our own fieldwork at the memorial, we found that people gave widely different interpretations of what the memorial meant and what people should and should not do there [34]. For instance, some participants’ interpretations of the memorial as a dignified place, representing the people being murdered during the Holocaust, led them to show strong negative opinions toward many behaviors seen at the site, such as taking selfies or jumping from one concrete block to another. Conversely, other participants’ more open interpretations seemed to make the site more compatible with these non-conventional behaviors. In general terms, memorials typically suggest a solemn and reflective attitude for visitors, which can easily conflict with others looking to get a good selfie. In protest to tourists’ behaviour, artist Shahak Shapira launched a project he called ‘Yolocaust,’ in which he pasted people’s picture poses at the memorial he found on social media, behind the scenery of the Nazi extermination camps instead, essentially making the abstractness and openness of the memorial concrete and figurative (see Section 3.2). The success of urban memorials has somewhat ironically transformed them into tourist attractions with a different set of ritual norms. This is especially true of both the 9/11 memorial and the MMJE.

### 3.9. Place of Burial

Anyone who has been in the presence of a corpse knows they have a special power over the human psyche. Anthropologists have carefully outlined the myriad of different rituals for handling bodies of the dead and their link to cosmological beliefs [48]. For example, Robert Hertz [49] described how the fate of the body, soul and bereaved were all tightly connected in Pacific islands’ funerary rituals. Typically, a double burial was practiced in which the ‘wet’ corpse was first buried, and at a later point, dug up as a ‘dry’ corpse and reburied in another location. This change symbolically marks the transition of the person’s soul as well as the survivors’ periods of mourning. When bodies are not properly attended to, this has implications for the status of the dead and the process of grief. This is often the case after violent events—Hertz himself was killed as a soldier in WWI. It was precisely the need to mourn the dead —many of them lying unidentified on the battlefields of the continent—that led to the idea of the unknown soldier after WWI. Because it could be anyone, regardless of their rank, many families were given a tangible site to grieve at. In contrast to this, the National Memorial for Peace and Justice (informally ‘the Lynching memorial’) in Montgomery (AL, USA) creates an unsettling atmosphere by hanging blocks that feel like bodies from the ceiling, which remind visitors of the gruesome death and unholy treatment of bodies as a pedagogical device (Figure 7).

While the material tangibility of corpses can facilitate the grieving process by helping relatives to acknowledge death, preventing them to be “left in a state of persistent and tormenting ambiguity, torn between hope and resignation” (p. 56, [1]), sometimes, the assumed presence of bodies is inevitable, especially on memorials built at the very site where traumatic events took place (see Section 3.10). This is the case of the memorials constructed on the Ground Zero sites in Hiroshima and New York [14], where problems arose in identifying the bodies, many of them “simply obliterated through modern technological violence” or merged with the rubble and debris (p. 313, [35]). Only half of the bodies of the 9/11 victims have been identified and of these, the vast majority are only fragments. Their status thus has parallels to the ‘disappeared’ in former dictatorships, such as Argentina and Chile, where their loss remains ambiguous in the absence of a body. At 9/11, the body fragments were placed in a room behind a wall in the memorial’s museum, which created several controversies about their access. In the Hiroshima Peace Park memorial, at the epicenter of the explosion, the Atomic Bomb Memorial Mound contains a vault with the ashes of nearly 70,000 victims that are still unclaimed either because their relatives have perished or because they have not yet been identified. According to the Hiroshima Peace Media Center, the remains of atomic bomb victims continue to be delivered to the memorial today. Thus, the site has become known as the “grave of Hiroshima” [50].

However, it is also the case that necropower [51], exercised over the corpses, purposively seeks to obliterate victims’ identities as a way to both erase any traces of human rights violations and mete out extra punishment to their relatives. As Ferrándiz and Robben point out, “the deliberate commingling of human remains in unmarked graves bewilders survivors and heightens the disorder, anxiety and division of the citizenry” (p. 1, [52]). This is the case of VF, whose crypt underneath the basilica hosts over 33,000 bodies from the Spanish Civil war, many of them unidentified and transferred without informing their families. As noted above (Section 3.7), this anonymous hodgepodge of corpses lying underneath is in stark contrast with the privileged and visible burial sites of Primo de Rivera and Franco by the basilica’s main altar. Franco’s recent exhumation, along with the first exhumations of those corpses moved there without their families’ permission—mostly from the Republican side—are constructing the path towards the “democratization” of the memorial and with it, the gradual healing of the still open wounds at both the individual and societal level.

### 3.10. Location and Surroundings

When a memorial is constructed at the site of the traumatic events, it is already affectively charged with the site’s history. The 9/11 memorial is designed to mark and contain the absence of the twin tours and the people who died there. The museum similarly focuses on mangled material objects from the site of destruction, reminding the visitor of the force of violence. There is also a tree highlighted on-site that survived the destruction and thus provides symbolism of resilience and hope. At the Hiroshima Peace Park (where an atomic bomb was dropped) they adopt a similar strategy, including the retention of the ruin of the Genbaku Dome. When new memorials are planned, there is often a discussion around their placement at the site of trauma or in another location. Chile’s Museum of Memory and Human Rights (including a memorial to the victims of Pinochet’s regime) was criticized for its placement outside the city center and with no connection to the repression (while many sites that were could have been used). A similar critique was made of MMJE, against the counter-argument that it should be placed in the middle of the capital where people cannot help but be confronted with it.

The surroundings of a site are equally important. VF was built near the town of El Escorial, where the Spanish royalty is buried, conferring the city’s symbolic value onto the site through its proximity. Moreover, by placing it on top of the Sierra, it creates a visual spectacle seen from miles around; in a literal and symbolic sense, Franco’s legacy is above the kings and queens. Similarly, VVM, 9/11 and MMJE are all central nodes in an urban network of memorial sites: VVM directly to the Washington Monument and Lincoln Memorial; 9/11 to smaller memorials located throughout NYC and New Jersey (not to mention, nationally); and MMJE to memorials commemorating other groups targeted by the Nazis, such as the Memorial to the Homosexuals Persecuted by the National Socialist Regime, and Sinti and Roma Memorial. One of the critiques of MMJE was in fact that it separates victims into groups as the Nazis had done. The National Memorial for Peace and Justice (Figure 7) goes a set further by inviting cities where lynching occurred to take a commemorative block with names of the victims into their community (the block is a duplicate of the one hanging in the memorial). Thus, visitors can see at the central memorial site, which cities have chosen to acknowledge the violent events and which continue to avoid doing so.

We can also consider the surroundings more directly as the physical space around the memorial site. Our participants at 9/11 frequently commented on the transition from the busy streets of NYC’s financial district into the peaceful park-like space of the memorial site. Being at the center of a major international city can also create tensions: both 9/11 and MMJE have become must-see tourist attractions, which encourage selfies for social media over reflection on the site’s meaning (Section 3.8). Finally, nature is a powerful frame in which to situate a memorial. Walter [53] argues that following the reformation, protestant countries took up nature as a symbol of healing in the absence of Catholic rituals of bereavement. That is why Protestant cemeteries tend to look like overgrown gardens or even forests, while Catholic cemeteries resemble walled cities. However, nature has since become a powerful symbol to both denominations. At VF, our participants emphasized the natural beauty of the surroundings (esp. those on the political right), even while the memorial itself takes the central stage. The World Memorial to the Pandemic (Figure 8), currently being built at the edge of Montevideo and the sea, is more innovative in refocusing our attention on the surrounding environment, such that the memorial itself becomes secondary. Similar to the 9/11 memorial, water is a key element evoking multisensory qualities (Section 3.5), but here its rhythms and flows are completely natural even as they are observed in the portal in the memorial’s center. The surroundings thus help to fortify the architects’ message that “humans are subordinate to nature and not the other way around,” thus de-centering humanity in the greater scheme of things. This echoes and affirms the findings of a recent commission report on ‘the value of death’ wherein the authors argue that “Climate change, the COVID-19 pandemic, environmental destruction and attitudes to death in high-income countries have similar roots—our delusion that we are in control of, and not part of, nature.” (p. 1, [54]).

## 4. Conclusions

Since ancient times, memorials have played a crucial role in practices of mourning and memory. While they contribute to the commemoration of a collective loss, they also enable individuals to make meaning of events and to find ways of personally connecting with the collective past. In conducting our research, we were fascinated by the power of architecture to set up fields of mnemonic experience for our participants. This is not to say that the way people experience memorial sites is determined by their material design or their intended meaning. As we have shown, people interpret and engage with memorials in manifold ways through different personal associations and memories that are aroused while exploring the site. Drawing on Gibson’s [55] notion of affordance, it could be argued that memorials’ design may facilitate, guide, invite, but also constrain, prevent, and forbid certain actions and ways of engaging with them. This implies a distributed agency between the individual and the possibilities of action offered by each memorial within a given sociocultural context [56]. These considerations deter us from attempting to assess a memorial site’s healing potential in general. The aim of this paper has been more modest: to develop a matrix for comparing material dimensions of memorials and visitors’ experience of them.

While the dimensions included in the matrix contain aspects that can be found in both traditional and counter-memorials, the results empirically support Stevens’ and Franck’s observation that counter-memorials “have given visitors more and more choices for how to engage with them, framing them as actors, not just viewers” (p. 29, [31]). For example, the vertical and affirmative style of traditional memorials, such as the VF–located on top of a mountain and endowed with a culturally recognizable functionality and meaning, tend to constrain the type of actions and rituals one can expect to perform, thus inhibiting spatial immersion with the site. This is in stark contrast with the more horizontal design of counter-memorials, such as the MMJE, whose abstract style plays on the ambiguity of its meaning. This invites multisensory exploration and greater mobility —beyond mere visual contemplation—along with an effort to provide these sites with meaning, whether through autobiographical memories or metaphorical associations. Engagement with counter-memorials is further afforded by a wide range of elements which furnish visitors’ experience with a more interactive and multisensory quality—tactile, sonorous, haptic, and kinesthetic [31]. Among them, we have seen the possibilities offered by elements, such as reflective surfaces (e.g., the evocative role of water in 11/9), the bell at the Hiroshima Memorial or the inscription of individual names inviting a physical connection with those we want to remember, an element that stood out in Watkins et al.’s study of VVM as a healing environment [15]. As Wasserman points out, these elements encourage and accommodate more personalized rituals, “helping visitors to feel they are a part of the memorial, while at the same time, preventing them from feeling helpless in the face of a tragic event” (p. 59, [8]). All in all, we could say that design features present in most counter-memorials seem to afford more personal ways of remembering in allowing each visitor to interpret and reckon with the collective past on her or his own terms. This brings us back to the idea of modern memorials understood as forums: in the place of temples revering a univocal version of the past, counter-memorials’ abstraction and minimalism “attempts to assign a singular architectonic form to unify disparate and competing memories” (p. 329, [32]), which we argue augments the healing potential of these places in pluralistic democratic societies.

However, as noted above, we should be cautious in drawing general conclusions about the effects of memorials’ design. As Watkins et al. remind us, “an effective design is not style-specific [but] context-specific” (p. 372, [15]). Experience of memorials results from the irreducible tension between individuals selectively engaging with the environment based on their personal history and the possibilities and constraints each site affords. In that regard, the experience will vary according to different factors, especially the profile of the visitor, with his or her unique life history and social position. As we approached the matrix from material design, we have not developed a comparison of different kinds of visitors and what they bring to the experience of memorial sites—given varying ages, political or ethnic identifications, or whether they are a first-time or repeat visitors. In our study of VF, for example, the mother in her 60s differed in her experience from other participants of the same family in associations with a Spanish church. Moreover, we intentionally did not attempt to recruit participants who had personally lost someone in the events being memorialized, which would have brought the person’s experience of traumatic loss to the fore. A different kind of matrix could and should be developed that starts from the dimensions of visitors, who are far from being a homogeneous group with similar needs. It would also be useful to consider more directly the collective memory that each site refers to, with its specific political considerations and closeness or remoteness to the past. The 9/11 memorial affords more personal memories of events (including the frequent mention of flashbulb memories) than MMJE and VF, simply in that it aims to represent and grieve a recent history. Material design, collective memory and visitor profile all work together to shape experience on-site. Finally, future studies might consider the digitalization of memorial sites—from QR codes to provide information about the memorial to using LED technology to display its core message [57]—as another design dimension different from the material features considered here. This brings in a new set of questions (e.g., about the alleged durability of digital technology versus its extreme frailty and dependence on energy supply) as well as novel forms of experiencing memorials by increasingly digitally engaged visitors.

## Figures and Tables

**Figure 1 ijerph-19-06711-f001:**
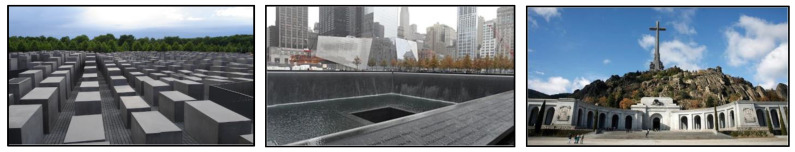
Memorial to the Murdered Jews of Europe (Berlin, 2005) (**left**), National 9/11 Memorial (NYC, 2011) (**center**), and Valley of the Fallen (El Escorial, 1959) (**right**).

**Figure 2 ijerph-19-06711-f002:**
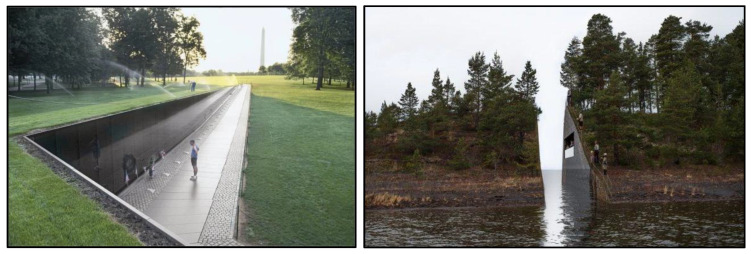
The Vietnam Veterans Memorial (Washington D.C., 1982) (**left**) and Utøya Memorial (Norway, never built) (**right**).

**Figure 3 ijerph-19-06711-f003:**
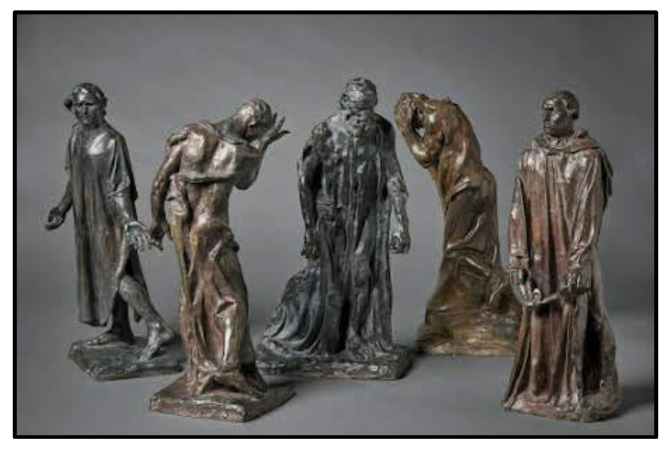
Rodin’s Burgher of Calais (1895).

**Figure 4 ijerph-19-06711-f004:**
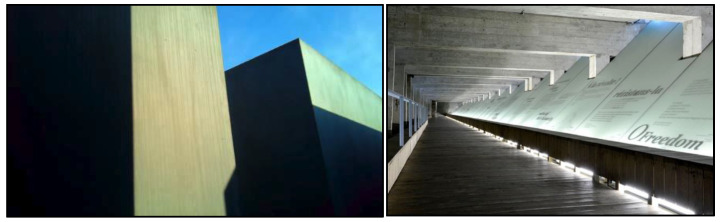
MMJE from the participant’s (subcam) (**left**) perspective and Memorial to the Abolition of Slavery (Nantes) (**right**).

**Figure 5 ijerph-19-06711-f005:**
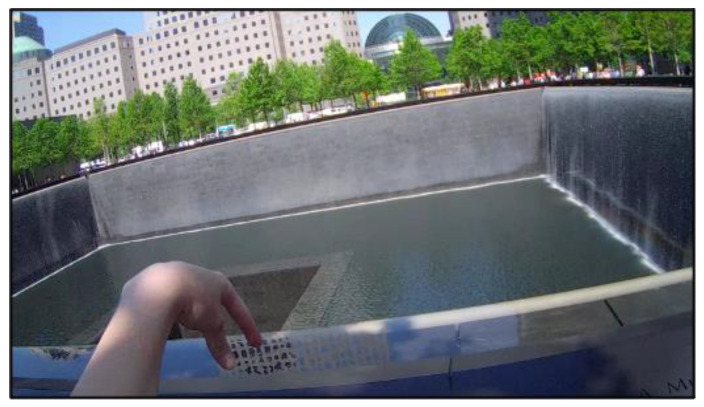
Participant reflecting at the 9/11 National Memorial.

**Figure 6 ijerph-19-06711-f006:**
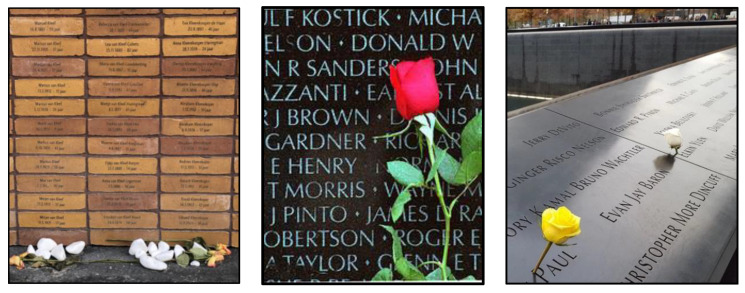
Individual names at the Holocaust Names Memorial (**left**), Vietnam Veterans Memorial (**center**), and 9/11 Memorial (**right**).

**Figure 7 ijerph-19-06711-f007:**
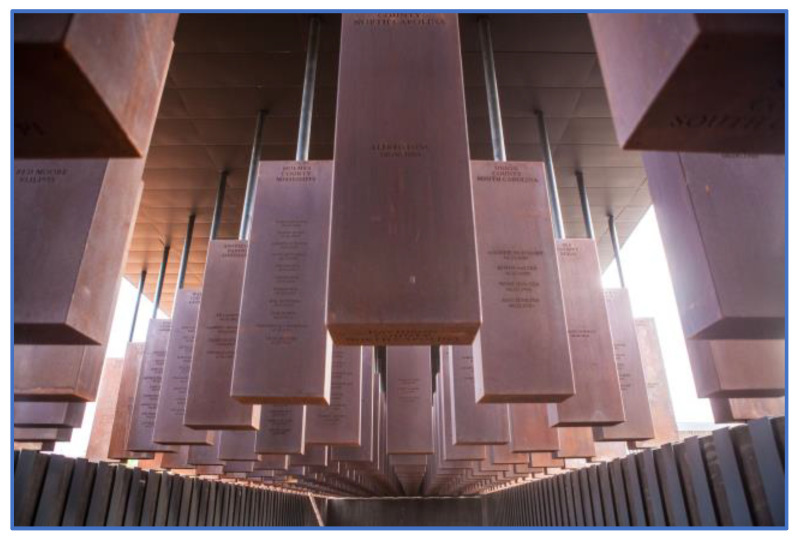
National Memorial for Peace and Justice (Montgomery, 2018).

**Figure 8 ijerph-19-06711-f008:**
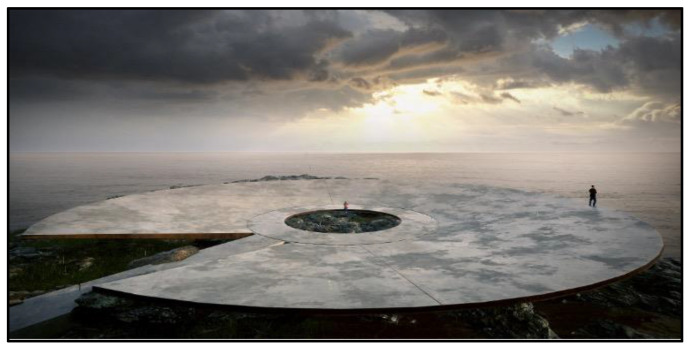
World Memorial to the Pandemic (planned, Montevideo).

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
