# Peer review of "Memorials as Healing Places: A Matrix for Bridging Material Design and Visitor Experience"

_ijerph, 2022, doi:10.3390/ijerph19116711_

Round 1

Reviewer 1 Report

I appreciated this article very much as I have visited some of these places that are discussed in this article. I also conduct pilgrimage programs in various places in Italy, and I can use the methodology presented in this article to help pilgrims visited the many secular and ecclesiastical sites in Italy.

Line 247  delete the the before England

Line 644  delete  (

Author Response

Thank you for your comments.  We are pleased the methodology and matrix will be of use in your own studies. We made the requested language changes.  The stray '(' was a result of automatic formatting when the paper was submitted.  Hopefully, it won't show up this time that way.  

Reviewer 2 Report

The article shows interesting research and, as the authors point out, much has been written about history, but little about how we appropriate historical events. The text provides important bibliographical references and lays the foundations for an interesting debate.
The article needs revision for several reasons:
It does not contextualize the case studies. It does not explain the reason for choosing these cases over many others. He talks about carrying out a survey that he does not explain or develop. It does not describe the profile of the respondents. It would be opportune, in the development of the material of the article, to explain the process of carrying out this survey.
It does not explain the research method used, simply conducting the survey. It would be opportune to contrast the method used.
The work briefly develops the materials and focuses on the results that seem to be broad conclusions and occupies the great extension of the text. It would be opportune to further develop a scientific methodology that allows knowing the data from which it is based in order to be able to carry out these conclusions.

Author Response

Thank you for your thoughtful review.  We agree that the research design was not sufficiently described. We have followed your advice to amend this by:

--further justifying our choice of case studies

--writing a long paragraph to properly contextualise the cases, including the builder's intentions and the sites' history.  

--further details on the profile of respondents and how the studies were carried out with them

In addition, we cleaned up the language throughout and added further details on the studies we conducted and links to other related research on the topic.

Reviewer 3 Report

The reviewer reserves the right to judge from the point of view of an art historian. It seems right to them that the Authors should become familiar with the fundamental literature in the field of research on the psycho-social impact of a work of art (i.a. R. Arnheim, E.H. Gombrich), which are also monuments. It establishes categories that seem to be relevant to this day in the analysis of the influence of a work of art. The terminology used by the Authors is imprecise, selective and sometimes colloquial.

The reviewer does not question the legitimacy of choosing basic examples for research. However, it is not sufficiently motivated.

It is also not presented how the formal and iconographical categories used for the analysis were selected. They are debatable and deviate from the customary ones (function, location, solidity and composition, material, figurative and inscriptional content, mobility, effects on other senses than sight).

The Results do not present a fundamental answer to the title problem – how and to what extent monuments heal or even affect the psyche by acting on the senses in selected formal and iconographic categories. This is what the reader would expect. Even if quantification is extremely difficult in this case, the qualitative results must be described.

In the Table in Conclusion, the record of visitors experiences is either obvious or very general.

Specific comments

The Authors do not use the terms “compositional axis”, “horizontal -” or “vertical composition”. In their place there are imprecise and undefined formulations “vertical -" and horizontal axis”. Translations of the meaning and method of perceiving horizontal and vertical systems are naïve.

Deriving abstract or abstracted symbols from the abstract art of the twentieth century is a misunderstanding.

It would be worth referring to research on the impact of specific solids and interiors for visitors.

Chapter 3.5. Multisensory Qualities is only limited to modest remarks about smell and touch.

It would be necessary to analyze the different types of materials and their impact - not only “reflective surfaces”.

It would be appropriate to assimilate the term “inscription”.

The wording “Presence of Bodies” seems awkward, maybe better to link the monument to “burial place”.

It is necessary:

to supplement in the introductory part information on research on the perception of spatial forms,

to explain the range of research,

to correct the accepted categories of formal and iconographic analysis,

to present the results of research – the feelings and emotions indicated by the visitors (if possible also in the quantitative record) assessing possible errors and distortions,

to draw conclusions

and to subject them to comparative analysis, with the results of similar research.

Author Response

Thank you for your comments and suggestions. We looked more deeply into Arnheim and Gombrich's work and indeed found it fascinating.  Arnheim's The Dynamics of Architectural Form was of particular interest.  That said, we did not restructure the paper according to categories of that field, which would have required writing a whole other paper.  

Instead, we have revised our categories to be more precise and where they aligned with formal art analysis we adopted those terms for our own purposes. We also stated more explicitly that we are coming from a memory and visitor studies perspective, and drawing mainly on that literature.  

The article's aim of the article is to develop a matrix of material design features that shape visitor experience, from our own studies and the available literature.  Realistically, it is beyond its score on the paper to make definitely conclusions as to their psychological and healing effect.  We have added further examples from our work and others though to fill it out and give further nuance to how the categories operate.  

We agree the table at the end was too general and not terribly helpful.  It has thus been removed.  Finally, throughout the article we have smoothed out the language, added context to the sites, further described our methodology and included other details about the studies.  

Round 2

Reviewer 3 Report

The authors have made improvements in many formal issues – i. a. literature, terminology. However, the main allegations have remained. No justification for the selection of examples has been made. The choice of the category of formal and iconographic analysis has still not been justified (they do not need to be changed). The problem of abstract characters in monuments has not been deepened. It has not yet been answered to what extent monuments heal, for example at list, as in the articles of N.Watkins. The article still requires the development of research results and rewriting part of the text.

Author Response

Responses to reviewer’s comments

No justification for the selection of examples has been made.

Justification is given in lines 99-103.  Our three case studies were the most prominent and widely discussed at the time.  They are also massive in scale, providing a site to explore people’s movement and a trajectory of experience. 

The choice of the category of formal and iconographic analysis has still not been justified (they do not need to be changed).

In lines 74-77 we argue that we are analysis is not based on formal and iconographic categories though they might overlap with them. Instead, we take our point of departure from memory and visitor studies. 

The categories first emerged through an inductive process of identifying dominant themes from our studies at each site. These were compared with each other with the memorial/counter memorial distinction in mind.  Finally we broadened out to include the wider literature and other sites discussed in it. In this way, it is a hybrid of an empirical and lit review (see lines 175-187)

The problem of abstract characters in monuments has not been deepened.

Perhaps we misunderstood the earlier comment here.  To respond to the comment about relation between art history and memorial abstraction we included lines 251-253.  And to strengthen the notion of the functions served and not served by abstract form we added lines 283-291 which illustrate the limitations of abstraction for certain kinds of healing.

It has not yet been answered to what extent monuments heal, for example at list, as in the articles of N.Watkins. The article still requires the development of research results and rewriting part of the text.

We have rewritten and extended the conclusion (lines 738-812) to bring the discussion together more adequately. We also came back to Watkins work on healing possibilities of memorials there. 

In the conclusion and above (e.g., in lines 39-42), we have also clarified our aim of developing a matrix for comparing material dimensions of memorials and visitor’s experience of them, rather than answering the question of to what extent memorials heal, which would have required a different approach.